# Chimeric Virus-Like Particles of Prawn Nodavirus Displaying Hepatitis B Virus Immunodominant Region: Biophysical Properties and Cytokine Response

**DOI:** 10.3390/ijms22041922

**Published:** 2021-02-15

**Authors:** Nathaniel Nyakaat Ninyio, Kok Lian Ho, Chean Yeah Yong, Hui Yee Chee, Muhajir Hamid, Hui Kian Ong, Abdul Razak Mariatulqabtiah, Wen Siang Tan

**Affiliations:** 1Department of Microbiology, Faculty of Biotechnology and Biomolecular Sciences, Universiti Putra Malaysia, Selangor 43400, Malaysia; nathanielninyio@kasu.edu.ng (N.N.N.); yongcheanyeah@hotmail.com (C.Y.Y.); muhajir@upm.edu.my (M.H.); 2Department of Microbiology, Faculty of Science, Kaduna State University, P.M.B. 2339, Tafawa Balewa Way, Kaduna 800241, Nigeria; 3Department of Pathology, Faculty of Medicine and Health Sciences, Universiti Putra Malaysia, Selangor 43400, Malaysia; klho@upm.edu.my (K.L.H.); onghk1991@gmail.com (H.K.O.); 4Department of Medical Microbiology, Faculty of Medicine and Health Sciences, Universiti Putra Malaysia, Selangor 43400, Malaysia; cheehy@upm.edu.my; 5Department of Cell and Molecular Biology, Faculty of Biotechnology and Biomolecular Sciences, Universiti Putra Malaysia, Selangor 43400, Malaysia; mariatulqabtiah@upm.edu.my; 6Laboratory of Vaccine and Biomolecules, Institute of Bioscience, Universiti Putra Malaysia, Selangor 43400, Malaysia

**Keywords:** prawn nodavirus, hepatitis B virus, virus-like particles, ‘a’ determinant, capsid protein, Sf9 cells, circular dichroism, cytokine, ELISA

## Abstract

Hepatitis B is a major global health challenge. In the absence of an effective treatment for the disease, hepatitis B vaccines provide protection against the viral infection. However, some individuals do not have positive immune responses after being vaccinated with the hepatitis B vaccines available in the market. Thus, it is important to develop a more protective vaccine. Previously, we showed that hepatitis B virus (HBV) ‘a’ determinant (aD) displayed on the prawn nodavirus capsid (Nc) and expressed in *Spodoptera frugiperda* (Sf9) cells (namely, Nc-aD-Sf9) self-assembled into virus-like particles (VLPs). Immunisation of BALB/c mice with the Nc-aD-Sf9 VLPs showed significant induction of humoral, cellular and memory B-cell immunity. In the present study, the biophysical properties of the Nc-aD-Sf9 VLPs were studied using dynamic light scattering (DLS) and circular dichroism (CD) spectroscopy. Enzyme-linked immunosorbent assay (ELISA) was used to determine the antigenicity of the Nc-aD-Sf9 VLPs, and multiplex ELISA was employed to quantify the cytokine response induced by the VLPs administered intramuscularly into BALB/c mice (*n* = 8). CD spectroscopy of Nc-aD-Sf9 VLPs showed that the secondary structure of the VLPs predominantly consisted of beta (β)-sheets (44.8%), and they were thermally stable up to ~52 °C. ELISA revealed that the aD epitope of the VLPs was significantly antigenic to anti-HBV surface antigen (HBsAg) antibodies. In addition, multiplex ELISA of serum samples from the vaccinated mice showed a significant induction (*p* < 0.001) of IFN-γ, IL-4, IL-5, IL-6, IL-10, and IL-12p70. This cytokine profile is indicative of natural killer cell, macrophage, dendritic cell and cytotoxic T-lymphocyte activities, which suggests a prophylactic innate and adaptive cellular immune response mediated by Nc-aD-Sf9 VLPs. Interestingly, Nc-aD-Sf9 induced a more robust release of the aforementioned cytokines than that of Nc-aD VLPs produced in *Escherichia coli* and a commercially used hepatitis B vaccine. Overall, Nc-aD-Sf9 VLPs are thermally stable and significantly antigenic, demonstrating their potential as an HBV vaccine candidate.

## 1. Introduction

More recent advances in vaccinology have extensively exploited virus-like particles (VLPs) as effective entities for use in vaccine delivery. Some viral antigens have been displayed on VLPs to enhance the immunogenicity of the antigens [1]. The resultant chimeric VLPs usually consist of multiple copies of a viral structural protein displaying the immunogenic epitope [1]. The VLPs of *Macrobrachium rosenbergii* nodavirus (*Mr*NV) have previously been employed as nanocarriers to display immunogenic viral epitopes of hepatitis B virus (HBV) and human influenza A virus (IAV) [1,2].

*Mr*NV is a non-enveloped RNA virus of the family *Nodaviridae* [3,4]. Its RNA genome consists of two single-stranded RNA molecules known as RNA1 and RNA2. The RNA1 (~3.1 kb) encodes the viral RNA-dependent RNA polymerase (RdRp), while the RNA2 (~1.2 kb) encodes the viral capsid protein (Nc) [4]. The Nc produced in *Escherichia coli* (*E. coli*) [5] and *Spodoptera frugiperda* (Sf9) cells [6] self-assembled into VLPs with diameters of ~30 nm and ~39 nm, respectively. The carboxyl-terminal (C-terminal) region of Nc has been reported to facilitate binding and internalisation of the virus into the host cells [4,7]. On the other hand, the arginine-rich amino-terminal (N-terminal) region of Nc is believed to facilitate RNA binding [8]. Furthermore, Nc derivatives fused with foreign viral epitopes at their C-terminal ends have been successfully produced in *E. coli*, and these chimeric proteins self-assembled into VLPs. These fusion epitopes such as the ectodomain of influenza A matrix 2 protein [2,9] and the ‘a’ determinant (aD) of HBV [1] were displayed on the surface of the Nc VLPs. In addition, these chimeric VLPs are believed to be potential vaccine candidates based on their ability to induce cellular and humoral immune responses against IAV and HBV, respectively.

Currently, there is no curative treatment for hepatitis B, and the viral infection is one of the most prevalent global public health problems [10]. HBV has infected about two billion people worldwide with about 240 million chronic HBV carriers [11], and about 800,000 yearly deaths are recorded globally due to complications arising from the viral infection [12]. In the absence of a cure, the available vaccines are the most reliable prophylactic measure against hepatitis B. However, vaccine failure reported in ~15% of vaccinees using the currently available hepatitis B vaccines shows that the vaccines are unable to protect some individuals such as the elderly, individuals with pre-existing health conditions and those with insensitivity to yeast-produced vaccines [13]. In certain cases, these vaccines do not elicit a robust or long-lasting immune response against certain HBV genotypes such as genotype C HBV [14]. This necessitates continuous developments of more effective hepatitis B vaccines.

To this end, chimeric VLPs harbouring the epitopes of the HBV surface antigen (HBsAg) have been successfully produced, characterised and shown to elicit the production of HBV-neutralising antibodies in mice [15,16,17]. A promising epitope for use in the development of chimeric VLP hepatitis B vaccines is the aD epitope [18,19]. Morphologically, the aD is a highly immunodominant region located from amino acid positions 100 to 169 within the conserved *S-*domain of HBsAg [11,15]. This immunodominant region has been identified as the major inducer of the humoral immunity, which leads to the production of cross-protective and neutralising antibodies against HBV [16,20,21]. Furthermore, the aD is also conserved across all HBV genotypes, a feature that makes it an ideal candidate for use in hepatitis B vaccine development [1,19].

Previously, we successfully produced Nc-aD in *E. coli* (Nc-aD-*E. coli*) [1] and Sf9 (Nc-aD-Sf9) cells **[19]**, in both cases the recombinant proteins self-assembled into VLPs. Immunisation of BALB/c mice revealed that Nc-aD-Sf9 VLPs induced a significantly higher humoral immune response compared to Nc-aD-*E. coli* VLPs and Engerix B (a commercial hepatitis B vaccine). Furthermore, we showed that Nc-aD-Sf9 induced a higher level of cytotoxic T-lymphocytes (CTLs) and a comparatively higher amount of natural killer (NK) cells and memory B cells than those of Nc-aD-*E. coli* and Engerix B [19]. To further support these findings, we deemed it necessary to analyse the biophysical properties of Nc-aD-Sf9 VLPs. Therefore, the objectives of this study were to analyse the biophysical properties of Nc-aD-Sf9 VLPs using dynamic light scattering (DLS) and circular dichroism (CD) spectroscopy, as well as to determine the cytokine profile elicited by the VLPs using multiplex ELISA. As a vaccine candidate, our findings showed that Nc-aD-Sf9 VLPs have the biophysical properties for use as a prospective hepatitis B vaccine.

## 2. Results

### 2.1. Generation of Recombinant Baculovirus Harbouring the MrNV Capsid Protein-HBV ‘a’ Determinant (Nc-aD) Coding Sequence

The coding sequence of Nc-aD was amplified and successfully inserted into the pFastBac HT C plasmid at *Nco*I *and Xho*I restriction endonuclease sites (Figure 1A). Recombinant bacmid DNA (Figure 1B) harbouring the *Nc-aD* gene insert was generated and transfected into Sf9 cells. Expression of Nc-aD in Sf9 cells (Nc-aD-Sf9) was analysed using western blotting, and a dominant protein band with a molecular mass of ~52 kDa was detected by anti-His monoclonal antibody (Figure 1C). The result showed a good agreement with the calculated molecular mass (51.3 kDa) of the recombinant Nc-aD-Sf9 protein (Figure 1D). The baculoviral stock generated was used for subsequent infection of Sf9 cells and scaled-up protein expression.

### 2.2. Expression, Purification and Quantification of the Nc-aD-Sf9 VLPs

Nc-aD-Sf9 VLPs from the culture supernatant and pellet of the Sf9 cells after 5 days of infection were purified using sucrose density gradient ultracentrifugation. SDS-PAGE analysis (Figure 2A) and western blotting (Figure 2B) showed the migration profile of the VLPs across the sucrose gradient with Nc-aD-Sf9 of ~52 kDa was detected from top (fraction 1) to bottom (fraction 18). Fractions 15 to 18 containing the ~52 kDa protein were pooled, dialysed, concentrated with a 30 kDa cut-off centrifugal concentrator, and analysed with SDS-PAGE (Figure 2C) and western blot analysis (Figure 2D). The Bradford assay was performed to measure the yield of the Nc-aD-Sf9 VLPs in the culture supernatant and cell lysate. One litre culture yielded ~9.1 mg of Nc-aD-Sf9 VLPs, which were distributed in the supernatant (~4.9 mg) and cell lysate (~4.2 mg). On the other hand, one litre culture of *E. coli* yielded ~2.4 mg of Nc-aD-*E. coli*. This demonstrates that Nc-aD VLPs produced in the insect cells were approximately 3.8-fold higher than those produced in bacteria.

### 2.3. Dynamic Light Scattering (DLS) Analysis

DLS analysis revealed that the averaged particulate size of the Nc-aD-Sf9 VLPs was 56.4 nm in diameter with a polydispersity index of 0.193 (Figure 3). However, the DLS analysis showed that majority of the VLPs (13.1%) were ~43.8 nm in diameter. The DLS results confirmed that the Nc-aD-Sf9 assembled into particulate VLPs.

### 2.4. Circular Dichroism (CD) Spectroscopy

CD spectrometry was performed to estimate the percentage of protein secondary structures of the Nc-aD VLPs. The CD analysis of Nc-aD-Sf9 revealed an α-helical spectrum (Figure 4A). A negative band was observed at 208 nm, and a positive band was observed at 192 nm. However, the CD analysis of Nc-aD-*E. coli* revealed predominantly a β-sheet spectrum. A negative band was observed at 216 nm, and a positive band was observed at 195 nm. Based on Reed and Reed’s reference [22], the estimated secondary structures of the Nc-aD-Sf9 VLPs at 20 °C are as follows: 16.1% α-helix, 44.8% β-sheet, 0.3% β-turn and 38.7% random coil, whereas the estimated secondary structures of the Nc-aD-*E. coli* VLPs at 20 °C are as follows: 7.4% α-helix, 54.1% β-sheet, 0% β-turn and 38.5% random coil. The Nc-aD-*E. coli* VLPs were found to contain more β-sheets and less α-helical structures than their Sf9-produced counterparts.

The estimated secondary structure changes of Nc-aD-Sf9 VLPs, at temperatures ranging from 20 to 70 °C, are shown in Table 1. The secondary structure measurements showed that as the β-sheet contents reduced across the temperatures, the α-helical contents increased. As shown in Table 1, the β-sheet was significantly shifted to α-helix and random coil at 50 °C and 60 °C. This is in good agreement with the temperature (51.9 °C) at which the Nc-aD-Sf9 VLPs started to denature (Figure 4B), and also the melting temperature (Tm) of Nc-aD-Sf9 VLPs (56.2 °C), as determined by thermal denaturation analysis. Nc-aD-*E. coli* VLPs had a higher melting temperature of 68.6 °C, however they started to denature at 45 °C (Figure 4B). This indicates that the Nc-aD-*E. coli* VLPs are less thermally stable than their Nc-aD-Sf9 counterparts, which started to denature at 51.9 °C.

### 2.5. Enzyme-Linked Immunosorbent Assay (ELISA) of Nc-aD VLPs

The antigenicity of the Nc-aD VLPs was analysed with ELISA. As shown in Figure 5, the Nc-aD-Sf9 and Nc-aD-*E. coli* VLPs were significantly more antigenic towards anti-HBsAg primary antibody than the negative controls including bovine serum albumin (BSA), nodavirus capsid produced in Sf9 (Nc-Sf9) and *E. coli* (Nc-*E. coli*). There was no significant difference between the antigenicity of Nc-aD-Sf9 and Nc-aD-*E. coli*.

### 2.6. Multiplex ELISA for Cytokine Quantification

To ascertain if the Nc-aD-Sf9 VLPs can induce cytokine production in in vivo systems, serum samples collected from each mouse injected with three doses of the Nc-aD VLPs were analysed using the multiplex ELISA. As shown in Figure 6, the Nc-aD VLPs induced the production of Th1 cytokine (IFN-γ), Th2 cytokines (IL-4, IL-5), macrophage and DC-secreted cytokines (IL-6 and IL-12p70), and IL-10 produced by macrophages, B-cells and DCs. This cytokine profile is indicative of macrophage, dendritic cell and cytotoxic T-lymphocyte activities, which suggests prophylactic innate and adaptive cellular immune responses mediated by Nc-aD-Sf9 VLPs.

## 3. Discussion

Sf9 cells have been explored as an efficient expression system for the production of VLPs displaying one or multiple foreign viral epitopes [22]. In contrast to *E. coli* expression systems, Sf9 cells have been shown to produce a higher yield of Nc, and they are also believed to be a better expression system for the production of VLPs for characterisation, structural and immunological studies [6,23,24]. Besides, VLPs of HBV [25], Nipah virus [26], flavivirus [27] and influenza A virus [28] have been successfully expressed in Sf9 cells using the baculovirus expression system. Thus, the Sf9 expression system presents good prospects for use in the production of Nc displaying foreign viral epitopes.

In the present study, the chimeric Nc-aD protein was expressed in Sf9 cells with the aid of the *Autographa californica* multiple nuclear polyhedrosis virus (AcMNPV) polyhedron promoter in the baculovirus expression system. The expressed Nc-aD protein was detected in the culture supernatant and cell lysate. The detected protein had a molecular mass of ~52 kDa, which corresponds well with the calculated molecular mass of 51.3 kDa. Furthermore, one litre culture of Sf9 cells yielded ~9.1 mg of Nc-aD, which is ~3.8-fold higher than that expressed in an *E. coli* expression system (~2.4 mg per litre). Although we did not quantify the yield of Nc-aD per solid mass of Sf9 and *E. coli* cells, our data suggest that the Sf9 expression system is a better choice for Nc-aD production in terms of protein yield. The result is consistent with that of the production of Nc alone using Sf9 cells [6], in which one litre of Sf9 cell culture yielded ~8.5 mg of Nc, while only ~1.5 mg/L of Nc was produced in *E. coli.* Furthermore, the estimated production cost of Nc VLPs was 19 USD/mg and 15 USD/mg in Sf9 and *E. coli* cells, respectively [6]. Although the production cost is more expensive, the overall protein quality, stability and immunogenicity of Nc VLPs produced in Sf9 cells are better than those produced in *E. coli* [6,19], rendering the insect cell a more preferable expression system.

DLS analysis showed that the Nc-aD-Sf9 assembled into particulate structures. As determined by DLS, the mean diameter of the Nc-aD-Sf9 VLPs was ~56.4 nm. The heterogeneous population of VLPs analysed via DLS showed that most of them (13.1%) were ~43.8 in diameter. However, the size of Nc-aD-Sf9 VLPs as observed by TEM was ~38 ± 17 nm [19], which is smaller than that determined by DLS analysis. The size difference could be due to the fact that DLS measures the size of particles in solution, and TEM involves size measurement of dried particles [29]. Thus, the larger size obtained via DLS is most likely due to the hydrodynamic shell formed around the particles, which gives them a larger hydrodynamic radius [29].

Far UV CD spectroscopy was employed to estimate the secondary structures of Nc-aD VLPs, which will provide a deeper insight into the stability of the chimeric VLPs. The result showed that the most abundant secondary structure in the VLPs was β-sheet, which constituted 44.8% and 54.1% for the Nc-aD-Sf9 and Nc-aD-*E. coli* VLPs, respectively. Both Nc-aD-Sf9 and Nc-aD-*E. coli* VLPs also have an α-helix composition of 16.1% and 7.4%, respectively. Previously, cryo-electron microscopic studies revealed that the Nc contained β-sheets in the capsid interior and the exterior protruding domains that form protruding spikes [24]. Recently, Chong et al. [30] demonstrated that the protruding domain of the Nc predominantly consisted of 67.9% β-sheets. The Nc and aD comprise 377 and 45 amino acid residues, respectively. In other words, the Nc makes up ~89% of the Nc-aD chimera. Therefore, we believe that majority of the β-sheet content of the Nc-aD is attributed to the high β-sheet content of its Nc component. Secondary structure prediction of the HBsAg showed that only the S-domain of HBsAg contains α-helices and β-sheets, which, together, are essential for the stability and folding of HBsAg components [31]. Because of the correlation between lower β-sheet content and stability of α-helical structures [32], it implies that the Nc-aD-Sf9 will be more stable than Nc-aD-*E. coli* because the former has a higher α-helix protein content.

The Nc-aD-*E. coli* had a higher melting temperature (~68.6 °C) than the Nc-aD-Sf9 (~56.2 °C). However, thermal denaturation analysis showed that the former started to denature at ~45 °C, while the latter started to denature at ~52 °C (Figure 4B). This shows that the Nc-aD-Sf9 VLPs are comparatively more thermally stable than Nc-aD-*E. coli* VLPs. We hypothesise that the difference in the α-helix content of the VLPs produced in Sf9 (16.1%) and *E. coli* (7.4%) accounts for the disparity in the initial denaturing temperatures of these VLPs. This is in line with Vogt and Argos’ [33] assertion that α-helical structures are more thermally stable than β-sheet and coil secondary structures. This further implies that, as a vaccine candidate, Nc-aD-Sf9 may withstand a wider range of handling temperatures without being denatured. As shown in Table 1, temperature interval protein secondary structure estimation of Nc-aD-Sf9 showed a complete denaturation of β-sheet at 60 °C suggesting a likely loss of protein structure and function.

The immunogenic epitope in chimeric VLPs needs to be displayed correctly on the surface of the particles. It is believed that epitopes displayed on the surface of VLPs have higher chances of interacting with components of the host’s innate immunity such as the surface pattern recognition receptors (PRRs) of DCs involved in viral sensing [34]. However, there is a likelihood of shielding of fused epitopes to occur during VLP assembly. This shielding could interfere with the interaction between these epitopes and components of the host’s immunity. A similar study highlighted the likelihood of partial shielding of terminally fused peptides during VLP assembly, and this shielding may impair peptide function [35]. As shown in Figure 5, the aD region of the Nc-aD VLPs was detected by anti-HBsAg antibodies in ELISA, indicating that the aD epitope was exposed on the surface of the Nc VLPs to allow for such an antigen-antibody interaction. It also suggests that the assembly of Nc-aD into VLPs did not suppress or mask the antigenicity of the aD epitope during VLP assembly. This is in good agreement with a previous study, in which TEM analysis of the immunogold stained chimeric Nc-aD-*E. coli* VLPs showed that the epitope fused to their C-terminal end was displayed on the surface of the VLP [1]. This is further supported by cryo-electron microscopic studies by Ho et al. [24,36], in which structural characterisation of Nc revealed its morphological suitability for the placement and display of foreign epitopes. These suggest that, as a display platform, the Nc is a suitable candidate to display immunogens in vaccine development.

An essential characteristic of vaccine candidates is their ability to induce the production of cytokines, which are crucial in the activation of the host’s innate immunity. The ability of Nc-aD-Sf9 to induce a specific/adaptive immune response has been previously discussed in Ninyio et al. [19]**.** As shown in Figure 6, Nc-aD-Sf9 and Nc-aD-*E. coli* elicited similar cytokine profiles although at different magnitudes. This suggests that the aD is responsible for the cytokine induction not the Nc carrier. Additionally, the difference in magnitude in the cytokine profiles of Nc-aD-Sf9 and Nc-aD-*E. coli* may be due to a better aD antigen presentation in Nc-aD-Sf9. Overall, Engerix B induced a Th2 biased immune response which is apparent in the lower titres of IFN-γ and IL-12 and the higher titres of IL-5 and IL-6. The production of cytokines (IFN-γ, IL-4, IL-5, IL-6, IL-10 and IL-12p70) are essential for conferring antiviral states on vaccinated subjects [37,38]. IL-12p70 induces T-helper cell 1 (Th1), which constitutes a major class of IFN-γ producing cells. Although IL-10 is expected to downregulate the production of Th1 cytokines such as IFN-γ, previously reported immunological analyses of Nc-aD-*E. coli* VLPs have shown significant production of IFN-γ in vaccinated mice even in the presence of IL-10 [1]**.** This could imply that the antagonistic property of IL-10 is short-acting, and may not be antagonistic against Th1 cytokines at week 9 when this multiplex-ELISA was performed.

During HBV infection, IFN-γ, which is also produced by NK cells, CTLs and macrophages, activates two virus-clearance pathways [37]. The first pathway mediates viral clearance by interfering with HBV genome replication and removing HBV nucleocapsid within hepatocytes [38]. The second pathway destabilises HBV RNA to interfere with the transcription of the HBV DNA [38].

On the other hand, IL-4 induces the differentiation of naïve T cells to Th2, and it has also been shown to suppress HBV replication in hepatocellular carcinoma cell line Hep3B [39]. The same study also demonstrated that IL-4 treatment decreased the amount of HBV DNA in infected hepatocellular carcinoma cells. Th2 induced by IL-4 produces IL-5, which, in turn, induces B cell growth and antibody secretion. This is in good agreement with the study we reported recently, in which Nc-aD-Sf9 induced a sustained antibody production and proliferation of memory B cells [19]. Induction of IL-6 and IL-10 by Nc-aD-Sf9 is indicative of macrophage activation. Although HBV has been shown to evade host innate immunity, it is reported that macrophages are an essential part of the innate immunity that is capable of HBV sensing [40]. Lang et al. [41] demonstrated the role of tissue macrophages (Kupffer cells) in inhibiting viral replication in hepatocytes. These Kupffer cells inhibit viral replication by sequestering the infectious virus particles, thereby interfering with their spread to neighbouring hepatocytes [40].

The release of cytokines mediated by Nc-aD-Sf9 VLPs may indicate DC activity in the immunised mice. These DCs are essential for the decrease in HBV titres when infection does occur. In a study performed by Shen et al. [42], DCs were co-cultured with the HepG2.2.15 human hepatoma cell line harbouring HBV DNA and covalently closed circular DNA (cccDNA). They showed that DC stimulation resulted in a significant decrease in intracellular HBV DNA and cccDNA levels. The DC stimulation also mediated a decrease in these HBV DNA and cccDNA released by the infected HepG2.2.15 cells into the culture supernatant. This suggests that Nc-aD-Sf9-mediated induction of DCs may be essential in HBV clearance in vivo.

Previously, we showed that Nc-aD-Sf9 VLPs induced a sustained antibody titre in immunised mice. ELISA revealed that the sera from these immunised mice recognised the aD epitope of HBV. Furthermore, we demonstrated that Nc-aD-Sf9 VLPs significantly induced the increase in NK cell, CTL and memory B cell populations [19]. In the present study, the cytokine profile elicited by Nc-aD-Sf9 VLPs suggests that the vaccine candidate could mediate HBV clearance by Th1- and Th2-regulated T-cell responses. The cytokines released are indicative of Nc-aD-Sf9-mediated activation of innate immunity, which is vital for viral sensing, and as the first line of defence against HBV infection before the adaptive immunity is fully activated.

## 4. Materials and Methods

### 4.1. Plasmid Construction

The pTrcHis-TARNA2 plasmid [1] was used as a template to amplify the coding sequence of Nc and HBV aD using the Phusion high fidelity DNA polymerase (Thermo Scientific, Waltham, MA, USA). A pair of primers, Nc-aD-F and Nc-aD-R (Table 2), were designed to amplify the coding sequence of Nc fused with aD (Nc-aD).

The initial denaturation was carried out at 98 °C for 30 s followed by 40 cycles of 98 °C (10 s), 72 °C (30 s) and 72 °C (25 s). The amplicons were purified with the QIAquick PCR purification kit (Qiagen, Hilden, Germany). The purified PCR products and plasmid vector were each digested with *Nco*I and *Xho*I.

The digested Nc-aD coding sequence was cloned into pFastBac HT C vector (Invitrogen, Carlsbad, CA, USA), which harbours the polyhedrin (P_H_) promoter of the AcMNPV. The *Nc-aD* gene was inserted at the *Nco*I and *Xho*I restriction sites of the linearised pFastBac HT C plasmid vector using T4 DNA ligase (Thermo Scientific, Waltham, MA, USA). The orientation of the insert was confirmed by PCR using a pair of primers Nc-aD/pFastBac HT C-F and Nc-aD/pFastBac HT C-R (Table 2), as well as restriction enzyme digestion using *Bgl*I and *Hind*III.

The recombinant plasmid was introduced into the MAX Efficiency DH10Bac *E. coli*, which harbours the bacmid DNA. Here, the mini-Tn7 element on the pFastBac HT C vector transposed the *Nc-aD* gene insert to the mini-attTn7 target site on the bacmid to generate a recombinant bacmid. Positive transformants were screened on Luria Bertani (LB) agar containing tetracycline (10 µg/mL), gentamicin (7 µg/mL), kanamycin (50 µg/mL), X-gal (100 µg/mL) and IPTG (40 µg/mL). The recombinant bacmid DNA was extracted using the alkaline lysis method and the presence of the *Nc-aD* gene insert was confirmed by PCR using primers pUC/M13-F and pUC/M13-R (Table 2) as previously described [6].

### 4.2. Cell Line and Transfection

Sf9 cells (ATCC^®®^ CRL-1711™, ATCC, Manassas, VA, USA) were grown to 80% confluence at 27 °C in Sf-900 III medium (Life Technologies, Carlsbad, CA, USA) supplemented with fetal bovine serum (FBS; 4%). The bacmid DNA containing the chimeric Nc-aD coding sequence was transfected into the Sf9 cells using the cationic lipid transfection method [43]. Briefly, in a 6-well plate, 8 × 10^5^ Sf9 cells were seeded per well and allowed to attach for 30 min. The Sf-900 III medium was aspirated, and replaced with Grace’s insect medium (2 mL; Life Technologies, Carlsbad, CA, USA). A transfection mixture consisting of Grace’s insect medium (200 µL), recombinant bacmid DNA (1 µg) and Lipofectamin 2000 (10 µL; Invitrogen, Carlsbad, CA, USA) was added to the Sf9 cells in a dropwise manner and incubated at 27 °C for 5 h. The medium was then replaced with fresh Sf-900 III medium (2 mL) supplemented with FBS (4%). The Sf9 cells were incubated at 27 °C for 4 days. The supernatant containing the baculovirus was collected and kept as stocks.

### 4.3. Production of Nc-aD in Sf9 Cells

Sf9 cells (2 × 10^6^ cells/mL) were infected with the baculoviral stock (10%), and incubated at 27 °C for 5 days until ~90% cells died. Cell debris and supernatant were separated by centrifugation at 200× *g* for 5 min. The cell pellet was resuspended in lysis buffer (15 mL; 77.4 mM Na_2_HPO_4_, 22.6 mM NaH_2_PO_4_, 1 mM phenylmethylsulfonyl fluoride, 0.4% Tween 20; pH 7.4) and sonicated three times at 200 W for 10 s with an interval of 2 min between cycles. The cell lysate and debris were separated by centrifugation at 12,000× *g* for 5 min at 4 °C. Then, proteins were precipitated from the culture supernatant and cell lysate using ammonium sulphate (50% saturation), and the precipitated proteins were resuspended in sodium phosphate buffer (2 mL; 77.4 mM Na_2_HPO_4_, 22.6 mM NaH_2_PO_4_; pH 7.4).

### 4.4. Purification of Chimeric Nc-aD-Sf9 VLPs with Sucrose Density Gradient Ultracentrifugation

The precipitated proteins were layered on a sucrose gradient solution, (10–40% (*w/v*)) and centrifuged at 150,000× *g* for 4.5 h at 4 °C using the SW 40 Ti rotor (Beckman Coulter, California, CA, USA). Sucrose solution was fractionated (500 µL), and the fractions containing the Nc-aD-Sf9, as determined by sodium dodecyl sulphate-polyacrylamide gel electrophoresis (SDS-PAGE) and western blotting, were pooled and dialysed. Dialysis was performed using a dialysis tube with a 10 kDa-molecular weight cut-off, in sodium phosphate buffer (3 L) for 24 h at 4 °C. The buffer was changed and dialysis continued for another 24 h. The dialysed protein was concentrated using a protein concentrator with a 30 kDa-molecular weight cut-off (Vivaspin tubes, Sartorius, Gottingen, Germany), and quantified using the Bradford assay [44].

### 4.5. Production of Nc-aD and Nc in E. coli, and Nc in Sf9

Expression and purification of Nc-aD produced in *E. coli* and Nc produced in *E. coli* and Sf9 cells were carried out as previously described [1,5,6]. Protein concentration was quantified using the Bradford assay [44].

### 4.6. SDS-PAGE and Western Blotting

The Nc-aD was mixed with 6X SDS-PAGE sample loading dye (100 mM Tris-HCl, pH 6.8, 20% (*v/v*) glycerol, 4% (*w/v*) SDS, 0.2% (*w/v*) bromophenol blue, 200 mM β-mercaptoethanol). The mixture was heated for 10 min, loaded in SDS polyacrylamide gel (12% (*w/v*)), and electrophoresed at 16 mA for 80 min. The gel was stained with staining solution (40% (*v/v*) methanol, 10% (*v/v*) acetic acid, 0.1% (*w/v*) Coomassie brilliant blue R-250) and destained with destaining solution (30% (*v/v*) methanol, 10% (*v/v*) acetic acid) until the protein bands became visible.

In Western blotting, proteins on a SDS-polyacrylamide gel were blotted onto a nitrocellulose membrane and blocked with 10% (*w/v*) skimmed milk (Anlene, Auckland, New Zealand) in TBS (50 mM Tris-HCl, 150 mM NaCl, pH 7.4) for 1 h. The membrane was washed three times with TBST buffer (TBS containing 0.1% (*v/v*) Tween 20). Anti-His monoclonal antibody (1:5000 dilution in TBS; Invitrogen, San Diego, CA, USA) was added, and incubated for 2 h. The membrane was washed three times in TBST buffer before incubating with anti-mouse antibody (1:5000 dilution in TBS; KPL, Milford, MA, USA) or anti-guinea pig antibody conjugated to alkaline phosphatase (1:5000 dilution; KPL, Milford, MA, USA) for 1 h. The membrane was then washed three times in TBST. Finally, colour development was performed by incubating the membrane with 5-bromo-4-chloro-3-indolyl phosphate (BCIP)/nitro blue tetrazolium (NBT).

### 4.7. Dynamic Light Scattering (DLS)

To determine the particle size and homogeneity of the Nc-aD VLPs produced by Sf9 cells, DLS was performed using the Zetasizer Nano ZS (Malvern, UK). The VLPs (in sodium phosphate buffer) were filtered with a syringe filter (0.2 µm) and transferred into the sample cell for analysis. The hydrodynamic radius (R_h_) of the suspended VLPs was measured by a 25 mW solid state laser at wavelength 780 nm at 20 °C.

### 4.8. Circular Dichroism (CD) Spectroscopy

The percentage of protein secondary structures present in the Nc-aD VLPs produced in Sf9 and *E. coli* was estimated using CD spectroscopy. Purified Nc-aD VLPs (10 µM; 400 µL) were loaded into a quartz cuvette (0.1 cm path length) and scanned with a circular dichroism (CD) spectrometer (JASCO J-815, Japan) at a speed of 100 nm/min at 20 °C. The CD spectrum was collected from wavelengths 190–240 nm. Measurements were done in triplicates. Collected data were analysed using the JASCO secondary structure estimation software [22,45]. The absorption reading for Nc-aD was corrected by subtracting the absorption reading of the sodium phosphate sample buffer. Measurements of the secondary structures at interval temperatures ranging from 20 to 100 °C were also performed from wavelengths 240 to 190 nm.

To determine the melting temperature (Tm) of Nc-aD VLPs, the samples were prepared as described above for the secondary structure estimation. The samples were subjected to thermal denaturation over a temperature gradient of 20–100 °C, and increment of temperature was maintained at a constant rate of 1 °C/min. Ellipticity and absorbance were recorded using the CD spectrometer (JASCO J-815, Japan), and the spectra were analysed using the JASCO Denaturation Analysis Programme.

### 4.9. Enzyme-Linked Immunosorbent Assay (ELISA)

Nc-aD VLPs and controls (positive and negative) with different concentrations (0.5, 1, 2, 5, 10, 15 and 20 µg/mL) were coated overnight in a 96-well microtiter plate at 4 °C. On the following day, the wells were washed three times with TBST, and blocked for 1 h with 200 µL of milk-diluent (KPL, USA). The wells were washed three times again with TBST, then anti-HBsAg antibody (1:2500 dilution in TBS; MP Biomedicals, Santa Ana, CA, USA; 100 µL) which interacts specifically with the aD, a component of the HBsAg, was added to each well and incubated for 1.5 h at room temperature. The wells were again, washed three times with TBST, and alkaline phosphatase-conjugated anti-guinea pig monoclonal antibody (1:5000 dilution in TBS; KPL, Milford, MA, USA; 100 µL) was added to each well and incubated for 1.5 h at room temperature. The wells were then washed with TBST before colour development by adding *p*-nitrophenyl phosphate (*p*-NPP) (50 µL per well), and incubated in the dark for 20 min. The absorbance was measured at 405 nm using the ELx800 microtiter plate reader (BioTek, Winooski, VT, USA).

### 4.10. In Vivo Studies on BALB/c Mice

Approval for the in vivo studies was granted by the Institutional Animal Care and Use Committee of Universiti Putra Malaysia (IACUC approval number R026/2018). Immunization of BALB/c mice with Nc-aD-Sf9, Nc-aD-*E. coli*, Engerix B, Nc-Sf9, Nc-*E. coli*, sodium phosphate buffer and alum, or sodium phosphate buffer only, was as described in Ninyio et al. [19]. The primary dose of the vaccine was administered to the mice on week 2. The first and second boosters were administered on weeks 5 and 8, respectively. On week 9, serum samples were collected as earlier described [19], and the serum samples from each immunisation group were used for cytokine quantification. This cytokine quantification will provide insights into the immunogenicity of the Nc-aD-Sf9 VLPs. The cytokine quantification was performed using multiplex ELISA.

### 4.11. Multiplex ELISA for Cytokine Quantification

Multiplex ELISA was performed using the mouse Th17 magnetic bead panel and kit (Merck, Darmstadt, Germany) according to the manufacturer’s protocol. Briefly, the 96-well plate was washed by adding the washing buffer (200 µL) into each of the wells. Following the washing step, each of the standards (25 µL) was added to the designated wells. The assay buffer (25 µL) was added to the sample wells and the background well. Then the serum matrix (25 µL) was added to the wells containing the background and standards. Furthermore, serum samples (25 µL) collected from the mice in each immunisation group was added into the designated wells. Analysis for each immunisation group was performed in triplicate wells. Then, the premixed beads were vortexed, and the beads (25 µL) were added to each well.

The 96-well plate was sealed and incubated overnight at 4 °C. Incubation was performed in the dark with gentle agitation. The contents of the wells were decanted, and the wells were washed twice with the washing buffer. Then, the detection antibodies (25 µL) were added to each well. The plate was sealed, and incubated in the dark with agitation for 1 h at room temperature. Then, Streptavidin-phycoerythrin (25 µL) was added to each well. Again, the plate was sealed, and incubated in the dark for 30 min at room temperature. The contents of each well were gently aspirated, and the wells were washed twice. Then, sheath fluid (150 µL) was added to each well followed by shaking for 5 min on a plate shaker to resuspend the beads. Finally, median fluorescent intensity was measured using a Luminex 200 multiplexing analyser (Luminex, TX, USA), and cytokine concentration was calculated using the xPONENT software.

### 4.12. Statistical Analysis

Variations in the antigenicity of the different antigens tested using sandwich ELISA, and variations in cytokine concentration were analysed with the one-way analysis of variance (ANOVA) using the Duncan’s multiple-range tests. *p* values less than 0.001 were considered as significant and those less than 0.0001 were considered as extremely significant. The statistical analysis was performed using the IBM SPSS statistics software (version 23).

## 5. Conclusions

Nc-aD VLPs produced in the insect Sf9 cells showed ~3.8-fold higher yield than those produced in *E. coli*, suggesting that Sf9 cells are a better expression system for Nc-aD VLPs. CD analysis revealed that the Nc-aD-Sf9 VLPs contained more α-helix (16.1%) than Nc-aD-*E. coli* (7.4%), which may be responsible for the higher thermal stability of Nc-aD-Sf9 (up to ~52 °C) in contrast to their *E. coli*-produced counterpart (stable up to 45 °C). ELISA revealed that the aD component of the Nc-aD VLPs was antigenic to anti-HBsAg antibodies indicating no suppression of aD’s antigenicity after fusion to Nc. In addition, multiplex ELISA of sera from the immunised mice revealed a higher induction of the cytokines IFN-γ, IL-4, IL-5, IL-6, IL-10 and IL-12p70 by Nc-aD-Sf9 VLPs as compared to Nc-aD-*E. coli* and Engerix B, which is indicative of NK cell, macrophage, and dendritic cell innate immune responses, as well as CTL activity. The stability and comparatively higher cytokine-mediated innate immune response induced by Nc-aD-Sf9 VLPs are indicative of a higher prophylactic efficacy than that induced by Nc-aD-*E.coli* VLPs and Engerix B.

## Figures and Tables

**Figure 1 ijms-22-01922-f001:**
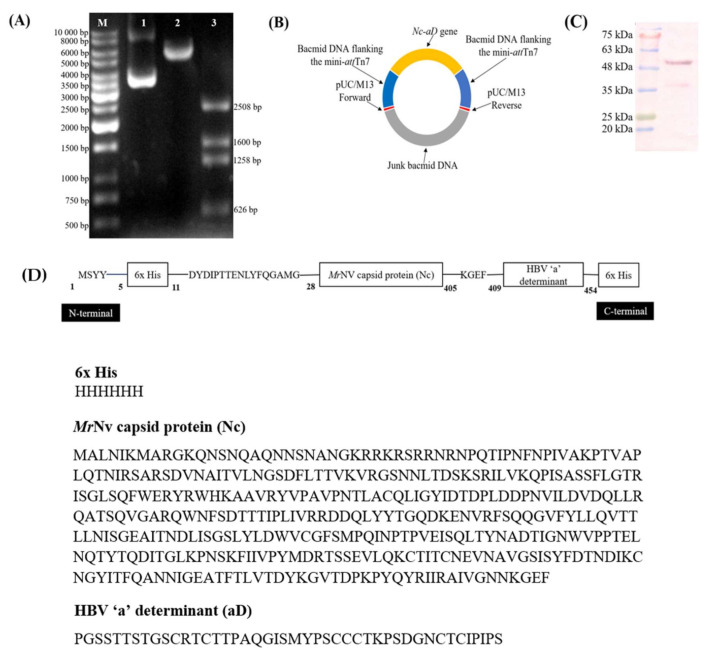
Construction of recombinant bacmid DNA harbouring the coding regions of nodavirus capsid (Nc) protein and HBV ‘a’ determinant (aD). (**A**) Restriction enzyme digestion of recombinant pFastBac HT C plasmid harbouring the *Nc-aD* gene insert. Lane 1: the undigested recombinant plasmid; lane 2: recombinant plasmid linearised with *Xho*I yielded an ~6 kb product; lane 3: the recombinant plasmid digested with *Bgl*I and *Hind*III yielded fragments with sizes 2508 bp, 1600 bp, 1258 bp and 626 bp. (**B**) A schematic representation of the recombinant bacmid DNA harbouring the *Nc-aD* gene insert. (**C**) Western blot of Sf9 cells transfected with the recombinant bacmid bearing the *Nc-aD* gene shows a protein band of ~52 kDa, corresponding to the expected size of the Nc-aD protein. (**D**) Primary structure of the chimeric protein consisting of *Mr*NV capsid protein (Nc) fused to HBV ‘a’ determinant (aD). The protein consists of the HBV aD fused to the C-terminal end of the Nc. The chimeric protein is flanked on both sides by 6x histidine (His) tags.

**Figure 2 ijms-22-01922-f002:**
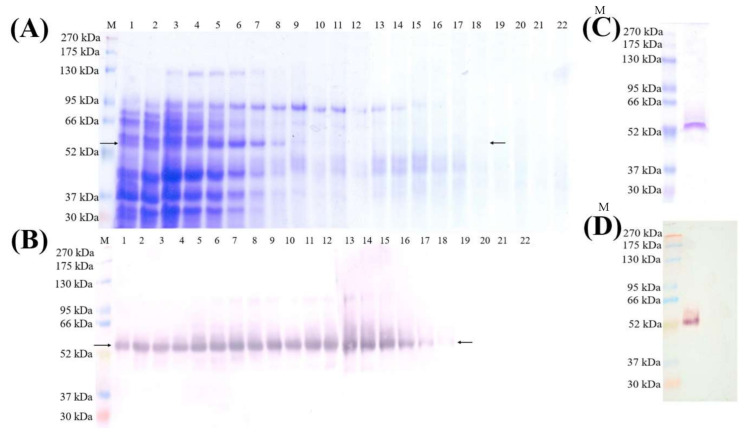
SDS-PAGE and western blot analysis of Nc-aD-Sf9 purified using sucrose density gradient ultracentrifugation. (**A**) SDS-PAGE and (**B**) Western blotting of cell lysate separated on 10–40% (*w/v*) sucrose density gradient ultracentrifugation with anti-His monoclonal antibody. This antibody was used because the chimeric protein contains His-tags at its N- and C-terminal ends. Arrows indicate the ~52 kDa Nc-aD-Sf9 protein band. Protein fractions in lanes 15 to 18 were pooled, dialysed and concentrated because they contained less impurities than samples in fractions 1 to 14. (**C**) SDS-PAGE of the concentrated Nc-aD-Sf9 protein (**D**) Western blotting of the concentrated Nc-aD protein with anti-His monoclonal antibody. Lane M: Molecular mass markers in kDa.

**Figure 3 ijms-22-01922-f003:**
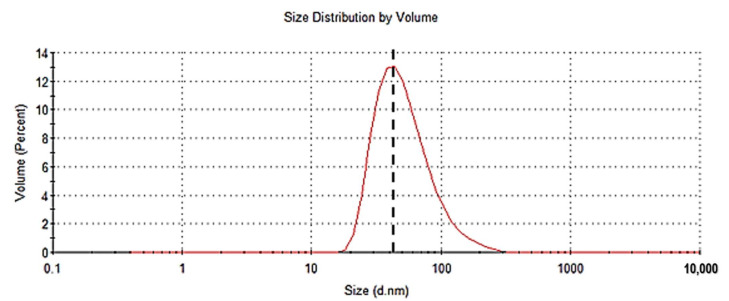
Dynamic light scattering (DLS) analysis of the Nc-aD-Sf9 VLPs. The vertical dotted line intersecting the red peak represents the mean diameter of the Nc-aD-Sf9 VLPs, 56.4 nm. DLS analysis showed that the largest population of the VLPs (13.1%) had a diameter of ~43.8 nm.

**Figure 4 ijms-22-01922-f004:**
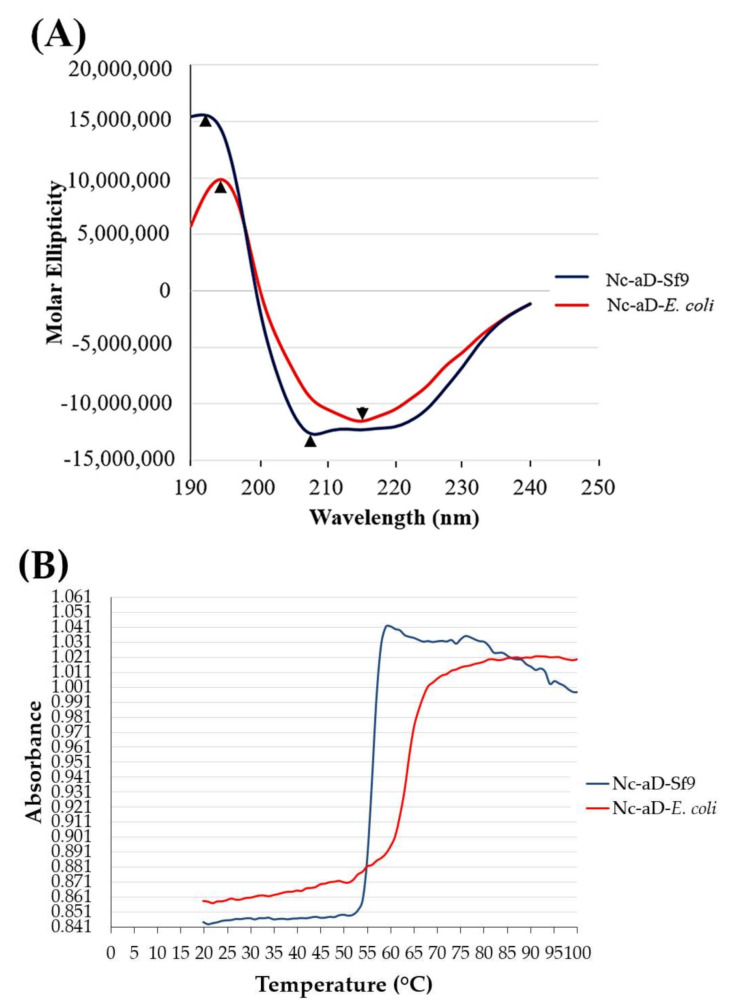
Circular dichroism (CD) spectra of the chimeric Nc-aD VLPs from wavelengths 240 nm to 190 nm. (**A**) The CD spectra showed the α-helix structure for Nc-aD-Sf9 as indicated by the negative band at wavelength 208 nm (arrow head) and a positive band at wavelength 192 nm (arrow head). The CD spectra showed a β-sheet structure for Nc-aD-*E. coli* as indicated by the negative band at wavelength 216 nm (arrow head) and a positive band at wavelength 195 nm (arrow head). (**B**) Thermal denaturation spectra for Nc-aD VLPs. The Nc-aD-Sf9 VLPs were stable from 20 to 51.9 °C (absorbance 0.844-0.849) with a Tm of 56.2 °C. Nc-aD-*E. coli* VLPs were stable from 20 to 45 °C (absorbance 0.859-0.870) with a Tm of 68.6 °C.

**Figure 5 ijms-22-01922-f005:**
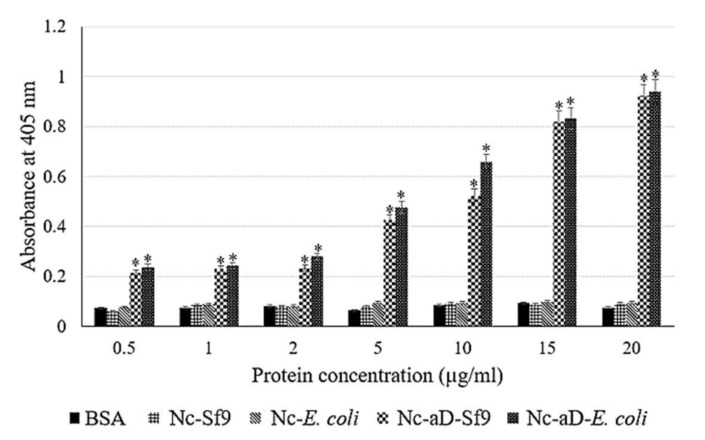
Antigenicity of the chimeric Nc-aD VLPs. The 96 well plates were coated with 0.5, 1, 2, 5, 10, 15 and 20 µg/mL of Nc-aD-Sf9, the negative controls (BSA, Nc-Sf9 and Nc-*E. coli*), and the positive control (Nc-aD-*E. coli*). The symbol * indicates *p* < 0.001 when compared to the BSA negative control. The error bars represent the mean (±) standard deviation. BSA: bovine serum albumin, Nc-Sf9: Nc produced in Sf9 cells, Nc-*E. coli*: Nc produced in *E. coli*, Nc-aD-Sf9: Nc-aD produced in Sf9 cells, and Nc-aD-*E. coli*: Nc-aD produced in *E. coli.*

**Figure 6 ijms-22-01922-f006:**
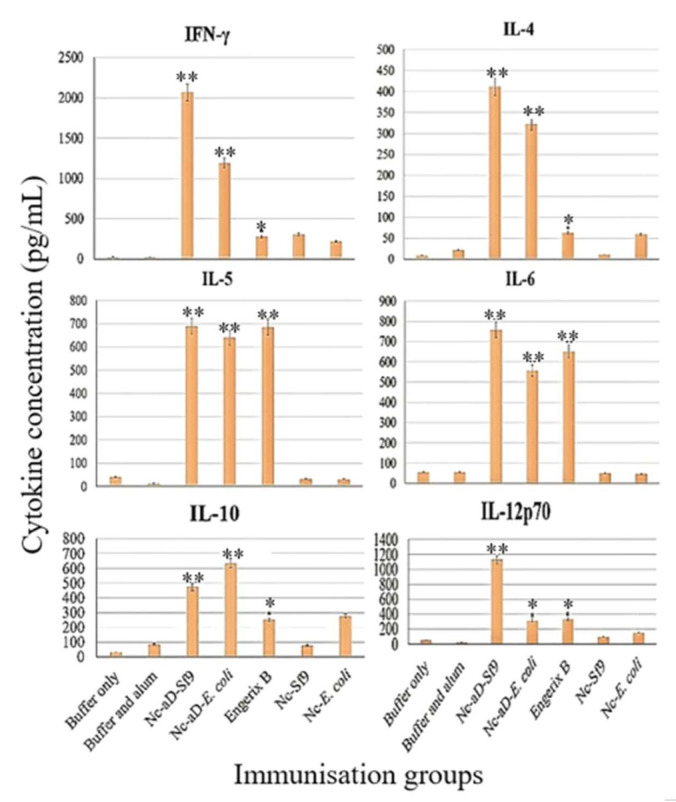
Cytokine quantification in sera of immunised mice. For each cytokine quantified, multiplex ELISA was performed using sera from mice in the test group (Nc-aD-Sf9), the positive control groups (Nc-aD-*E. coli* and Engerix B) and the negative control groups (Buffer only, Buffer and alum, Nc-Sf9 and Nc-*E. coli*). The symbols * and ** indicate *p* < 0.001 and *p* < 0.0001, respectively, when cytokine concentrations from the test group (Nc-aD-Sf9) and positive control groups (Nc-aD-*E. coli* and Engerix B) are compared to that of the ‘Buffer only’ negative control group. The error bars represent the mean (±) standard deviation. Nc-aD-Sf9: Nc-aD produced in Sf9 cells, Nc-aD-*E. coli*: Nc-aD produced in *E. coli*, Nc-Sf9: Nc produced in Sf9 cells, and Nc-*E. coli*: Nc produced in *E. coli*.

**Table 1 ijms-22-01922-t001:** Temperature interval protein secondary structure estimation for the Nc-aD-Sf9 VLPs.

Temperature (°C)	α-Helix (%)	β-Sheet (%)	β-Turn (%)	Random Coil (%)	Total (%)
20.0	17.9	43.6	5.7	32.8	100.0
30.0	20.6	48.8	1.9	28.7	100.0
40.0	26.1	40.8	2.4	30.7	100.0
50.0	33.5	5.9	8.0	52.6	100.0
60.0	52.5	0.0	2.8	44.6	100.0
70.0	25.2	0.3	28.0	46.4	100.0

**Table 2 ijms-22-01922-t002:** Primers used for the amplification and detection of *Nc-aD* gene insert.

Primer	Primer Sequences	T_a_ (°C)	Function
Nc-aD-F Nc-aD-R	5′-GGGGGCCATGGGGATGGCCCTTAACATCAAG-3′ 5′-CCCCCTCGAGTTAATGATGATGATGATGATGG-3′	72	Amplification of *Nc-aD* gene from recombinant pTrcHis-TARNA2 plasmid
Nc-aD/pFastBac HT C-F Nc-aD/pFastBac HT C-R	5′-CCATGGCGTACTACCATCAC-3′ 5′-GTAAAACCTCTACAAATGTGG-3′	59	To detect the *Nc-aD* gene in the recombinant pFastBac HT C vector
pUC/M13-F pUC/M13-R	5′-CCCAGTCACGACGTTGTAAAACG-3′ 5′-AGCGGATAACAATTTCACACAGG-3′	55	To detect the *Nc-aD* gene in the recombinant bacmid DNA prior to transfection into Sf9 cells

The underlined nucleotide sequences in the forward and reverse primers are the *Nco*I and *Xho*I restriction sites, respectively. T_a_ indicates the annealing temperatures of the primers.

## Data Availability

The data presented in this study are available in article.

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
