# Peer review of "Chimeric Virus-Like Particles of Prawn Nodavirus Displaying Hepatitis B Virus Immunodominant Region: Biophysical Properties and Cytokine Response"

_ijms, 2021, doi:10.3390/ijms22041922_

Round 1
Reviewer 1 Report
In this manuscript, Ninyio NN et al. analyze the chimeric prawn nodavirus VLP's biophysical properties harbouring HBV "a" determinant. The authors also measure the cytokine release in immunized Balb/c mice. Although the manuscript is very long, redundant in many places, the results obtained are very few.
Major comments:
1) Lots of details about the generation of the recombinant bacmid DNA, including some control digestions (fig. 1A) but we couldn't find out where aD is positioned relative to NC and where the His tag is.
2) What is the difference between the expression and purification of Nc-aD-Sf9 VLPs obtained previously [19] and those described in the manuscript? If there is no difference, why is this considered a result?
3) In figure 2 B and D, the authors used anti-His tag antibodies instead of anti-HBsAg antibody, without explaining why. Also, it is not clear why only fractions 15-18 were pooled when a lot of protein is found in the other fractions as well.
4) The antigenic loop of HBV S protein accommodate a partially occupied glycosylation site (146) and several cysteines that form disulphide bridges. The authors did not discuss this or test whether antisera from the immunized mice can recognize or even neutralize HBV.
Minor comments:
1) There are other ways to represent statistical significance in a chart, without listing the entire alphabet (fig. 5 and table 2).
2) It is not very correct to compare the yield of antigen production per litre of cell culture; it would be more accurate to refer to the solid mass of Sf9 or E. coli cells.
Author Response
Please see the attachment for the point-by-point responses to Reviewer 1 comments.

Reviewer 2 Report
This study provides potentially interesting information. However, there are some flaws need to be addressed, as followings.
Comments
- Abstract, page 1, line 35: IL-10 is known as immunosuppressive cytokine, and it should antagonize IFN-gamma too. How the authors can explain the induction of IL-10 and IFN-g. Moreover, how the authors can justify induction of IL6 and IL10 as prophylactic response of a proposed vaccine candidate?
What about the expression status of IL10 in Engerix B?
- Introduction, page 3, line 99:”significantly” is better to be “highly”.
- Results, page 4, line 134: “the culture supernatant and lysate…” should be “the culture supernatant and cell lysate…”
- Results, page 4, line 137: “Nc-aD VLPs produced in the insect cells was approximately 3.8 folds higher…” should be “Nc-aD VLPs produced in the insect cells were approximately 3.8 folds higher…”
- Results, page 7, line 209: “innate and adaptive cellular immune response…” should be “innate and adaptive cellular immune responses…”
- Results, page 7; Fig.5, statistical analysis in this result is not clear.
- Page 10, line 276-7, Do the authors consider this kind of Ag-Ab interaction is enough to be a vaccine candidate? Did the authors perform virus neutralization assay using the proposed vaccine candidate, if no, it is very important to perform to confirm the suitability as a vaccine candidate.
Author Response
Please see the attachment for point-by-point responses to Reviewer 2 comments.

Reviewer 3 Report
The authors described an interesting study which reports the use of prawn nodavirus virus like particles (VLPs) as a vaccine delivery system for the presentation of the hepatitis B virus (HBV) ‘a’ determinant antigen. The current study builds on previously published work where the nodavirus VLP with the ‘a’ determinant (referred to as Nc-aD) was expressed in Escherichia coli by expressing the VLP construct in an Sf9 insect cell system. The authors report the characterisation of the Nc-aD-Sf9 VLP biophysical properties. They also report the evaluation of the immunogenicity of the VLP in mice.
The abstract requires a major rewrite. A good abstract should provide some context/background to the study. How the study was done. What the key/major findings of the study were (not all just key ones). What the final conclusions were. What the results of the study mean in the context of the relevant literature. This provides the prospective reader with a full appreciation of the study. The current abstract really just summarises the paper.
There needs to be a much clearer explanation of the relationship of this manuscript and the recently published manuscript Ninyio et al. (10.3390/vaccines8020275). It would seem that additional analyses of samples from the same immunisation experiment are reported in this manuscript. Is this correct? While this is completely acceptable, where justified, any relationship must be absolutely clear and justifiable. For clarity, text such as “the results of memory responses and circulation cell phenotypes are reported elsewhere (cite the reference). Justification would be typically along the lines of using the collected samples for different types of analyses. I am not convinced the current manuscript, in its current, form does this. While reading the manuscript, I was wondering why it was focused on innate immune responses and not specific/adaptive responses. Is it reasonable to look at innate responses so long after immunisation? This for the authors to justify. Is it necessary to present Western blotting of the hybrid nodavirus VLP in this manuscript when a similar analysis was published previously?
Other comments:
Lines 96-106 This text basically summarises the results of the manuscript. I would suggest it be rewritten to provide the aims and/or objectives of the study.
Line 47 suggest replacing “design” with “delivery”
Line 108 Section 2.1 – Are these new results? The authors stated previously that they have already reported the capacity of the Nc-aD to self-assembly into VLPs in an insect expression system (line 86).
Line 127 Section 2.2 – see the previous comment. Has DLS analysis been conducted on the bacterial expressed VLP?
Line 188 Section 2.5 – it would be useful to mention what primary antibody(s) were used in this analysis.
Line 211 I think the presentation of this data requires improvement. Did the authors consider presenting this data graphically, on the basis of individual cytokines? On the basis of the footnote, it seems as though there have been blanket cross data comparisons made, is this correct?
If so, the authors should explain why this was done. For example, why would one compare the level of IL-4 expression to the level of IL-10 expression for a given treatment? Apologies if I have incorrect interpreted this. I think graphing this data would help.
Line 225-234 – While the insect system provides a higher yield of the VLP would this be the only consideration in producing the antigen for a vaccine? Would a 3.8 fold increase in yield account for the relative costs of these systems?
Line 425 – Is the anti-HBsAg antibody specific to the ‘a’ determinant? If so, this should be specifically stated.
Author Response
Please see the attachment for the point-by-point responses to Reviewer 3 comments.

Round 2
Reviewer 1 Report
Comments on the author's response:
1) "Our previous publication describes briefly the Nc-aD VLPs expressed in Sf9 cells. In the present manuscript, we decribed in depth the details of the baculovirus expression process." So, your previous publication was incomplete, and you give the details of the baculovirus expression in this manuscript? Anyway, the sucrose density gradient ultracentrifugation usually is not sufficient for highly VLP purification. What is the purity of the injected antigen? You consider it to be 100% from the coomassie stained gel?
2) "only fractions 15-18 were pooled because they had lesser impurities than the other fractions." All proteins are less in those fractions, some of them below the detection limit; this does not necessarily mean that the antigen is purer.
3) "This antibody was used because the chimeric protein contains His-tags at its N- and C- terminal ends". The fact that the antigen has two His-tags is not a reasonable explanation of why you do not use an anti-HBs antibody. If anti-HBs antibodies do not recognize Nc-aD-Sf9 in the Western blot, you should mention it. But then there is a problem with the ELISA experiment.
Author Response

(The authors gave the same response as above.)

Reviewer 2 Report
Authors adequately addressed the questions raised by reviewers.
Author Response
Many thanks for the comment.
Reviewer 3 Report
The authors have done an excellent job in the revision of their manuscript. They have addressed all of the comments and suggestions from the review of the submitted version of the manuscript.
I have no further comments.
Author Response
Many thanks for the comments.
Round 3
Reviewer 1 Report
No more comments.